# Severe Abnormalities of Lens Epithelial Cells in Exfoliation Syndrome: A Transmission Electron Microscopy Study of Patients with Age-Related Cataract

**DOI:** 10.3390/medicina55060235

**Published:** 2019-05-31

**Authors:** Konstantina Ν. Sorkou, Maria Eleni Manthou, Soultana Meditskou, Nikolaos Ziakas, Konstantinos T. Tsaousis, Ioannis T. Tsinopoulos

**Affiliations:** 1Laboratory of Histology and Embryology, Medical School, Aristotle University of Thessaloniki, University Campus, 54124 Thessaloniki, Greece; meditskou@gmail.com; 22nd Department of Ophthalmology, Medical School, Papageorgiou Hospital, Agiou Pavlou 76, 56429, Aristotle University of Thessaloniki, 54124 Thessaloniki, Greece; nikolasziakas@gmail.com (N.Z.); kttsaousis@yahoo.gr (K.T.T.); itsinop@med.auth.gr (I.T.T.)

**Keywords:** exfoliation syndrome, age-related cataract, senile cataract, anterior lens capsule, lens epithelial cells, electron microscopy

## Abstract

*Background and objectives*: The aim of this study was to examine via electron microscopy the lens epithelial cells in age-related cataracts and compare the findings between patients with and without exfoliation syndrome, in the Greek population. *Materials and Methods*: Twenty-one patients with age-related cataracts, older than 60 years, were included in the study. Eleven of them also suffered from exfoliation syndrome. Anterior lens capsules, obtained during phacoemulsification, were examined with a transmission electron microscope. *Results*: In all cases, ultrastructural features of diffuse intracellular and extracellular oedema were noticed to a varying degree and transparent vacuoles were detected. Often, there was more than one layer of cells, giving the impression that healthier cells tried to cover neighboring cells presenting extensive damage. Commonly, cells lost their regular shape and appeared with expanded nuclei carrying dense granules. Apoptotic cells were also detected. The epithelial cells frequently were completely destroyed or absent, exhibiting loose connections amongst them or with the basement membrane. In exfoliation syndrome (XFS) patients the alterations were more severe. Additionally, the lens epithelial cells (LECs) apical cell membrane appeared with varying distances from the basement membrane, due to different cell “heights”, creating an irregular margin of the epithelium (*p* < 0.05). *Conclusion*: Transmission electron microscope (TEM) examination revealed ultrastructural abnormalities in all patients’ lens epithelia, more extended and more frequently observed in XFS group. In all cases, the lesions were comparable to those described in severe pathologies, all of which were excluded from the study. Environmental factors such as increased ultraviolet B (UVB) radiation exposure in Mediterranean countries, genetic factors, epigenetic factors, or all of them, could contribute to these alterations. Further epidemiological and molecular biology research is needed, so as to justify these results.

## 1. Introduction

Ageing is the most common cause of cataract, but other various risk factors may also be implicated in its etiopathogenesis, such as genetic, traumatic, metabolic, toxic agents or radiation. Senile cataracts have become gradually more severe and frequent in the elderly and is the most common cause of blindness worldwide [1]. Cataract formation seems to be associated with the structural organization and function of the lens epithelium, which is the regulating barrier between aqueous humour and lens fibers [2]. Metabolically, the epithelium is the most active compartment of the ocular lens. It regulates the ion concentration and water accumulation, protecting the lens inside. Any factor having an impact on transport processes or morphology of the lens epithelial cells (LECs) may disturb lens transparency [3,4].

Considering the important functions attributed to the lens epithelium, damage to it has been an area of interest for researchers. It is generally accepted that LECs have typical epithelial morphology. They are very closely arranged in a single layer. They are cuboidal with round nuclei and have smooth surfaces under the capsule and over the underlying lens fibers [5]. Only a few studies have focused exclusively on the description of lens epithelium in senile cataracts with use of a transmission electron microscope (TEM), and none of them wererecent [6,7]. Mostly, anterior lens capsules of age-related cataract patients constitute the control group in TEM studies, which investigate other pathologies [8,9,10]. In these studies, mild abnormalities of LECs are described, such as inter- or intracellular vacuoles.

We recently studied anterior lens capsules (aLCs), composed of the lens epithelium and the basement membrane, from patients with senile cataracts and exfoliation syndrome (XFS) via TEM, for the first time in Greek or Mediterranean populations. The study focused on the findings in the subepithelial region of aLC, towards the lens fibers. We reported the presence of a new, not previously described, unbound material, consisting of electron-dense microgranules or larger formations, on the apical side of the lens epithelium in the XFS group [11].

Exfoliation syndrome is considered the most common identifiable cause of glaucoma worldwide [12] and is associated with an increased incidence of surgical complications in cataracts [13,14,15]. It is an age-related disorder, recognized biomicroscopically from the presence of white deposits on the anterior lens surface. XFS is characterized by the production and chronic accumulation of an abnormal fibrillar material in many ocular and extraocular tissues [16]. The prevalence of XFS in Greece, as recorded in epidemiological studies, is relatively high compared to other countries [17,18,19,20,21,22,23].

The present study focuses on the characteristics of lens epithelial cells, comparing the findings between senile cataract patients with and without XFS, as well as with documented electron microscopic findings of other population groups.

## 2. Materials and Methods

Twenty-one patients with age-related cataracts who visited the 2nd Department of Ophthalmology at Papageorgiou University Hospital of Aristotle University in Thessaloniki, Greece, were included in the study. The patients were older than 60 years and eleven out of them also suffered from XFS. Exclusion criteria for the patients’ recruitment were specific radiation exposure, uveitis, retinitis pigmentosa, Wilson’s disease, Alport syndrome and non-senile or intumescent white cataracts. These are all conditions that have been described to cause degenerative alterations to the lens epithelium [8,24,25,26,27,28,29,30,31,32,33]. The nature of the study, which follows the tenets of the Declaration of Helsinki, and all the scheduled procedures, were thoroughly explained to all patients. Before the operation the patients were asked to sign informed consent forms. The Ethical Committee of the Aristotle University of Thessaloniki and the Hellenic Data Protection Authority approved the study (Number of approval: ΓΝ/ΕΞ/1445-1/27.04.2015).

One surgeon from our research group (IT) performed uneventful phacoemulsification at Papageorgiou University Hospital in Thessaloniki on all of the patients included in the study. After continuous curvilinear capsulorrhexis he carefully removed a 5–5.5-mm circular portion of the central aLC from each patient. All specimens were immediately sunk into a 3% glutaraldehyde solution in neutral buffer and were then transported at the Laboratory of Histology and Embryology, Aristotle University of Thessaloniki. This is where specimens’ preparation was completed and visualization on the electron microscope was performed.

All specimens were left in the glutaraldehyde solution for 90 min and were then post fixated in a 2% OsO4 solution for one hour. Dehydration in increasing concentrations of ethanol followed, and finally the capsules were embedded in Epon 812. Firstly, 1–3 μm semi-thin sections were obtained from the center of the capsule, perpendicular to the specimen’s plane. The sections were stained with 1% cyane toluidine and were observed under a light microscope. When the area of greatest interest was chosen, golden, ultrathin sections (600–800 nm) were acquired, which were then stained with uranyl-acetate and lead-citrate. Finally, the sections were examined using a JEM-1011 transmission electron microscope (JEOL, Inc., Peabody, MA, USA), made in Japan. For each aLC, six different regions of at least five cells were used from various parts of the specimen for our recordings. When proceeding in a deeper part of the specimen to cut the next sections, a step of at least 250 μm was applied.

The Shapiro-Wilk test was used for examining the normality of the data. The Student’s *t*-test, Chi-square test and Fisher’s Exact test were chosen for comparisons between groups, as appropriate. The level of statistical significance was defined at 5%. All tests were two-tailed. IBM SPSS Statistics, Vesrion 25.0 (IBM Corp., Armonk, NY, USA) was used for the statistical analysis.

## 3. Results

The demographic and clinical data of the cataract patients included in the study, with and without XFS, are shown in Table 1 and Table 2. The patients without XFS are referred to as the age-related cataract (ARC) group. The TEM observations of lens epithelial cells were more extended and either more frequently or exclusively observed in patients with XFS (Table 3).

Changes of the lens epithelium, which may be characterized as degenerative, were observed in all specimens. The examination of lens epithelia revealed features of diffuse intracellular oedema, along with extracellular oedema in varying degrees (Figure 1, Figure 2, Figure 3 and Figure 4). Transparent vacuoles of various sizes were detected between the cells and between cells and the basic membrane, in all cases (Figure 1, Figure 3D and Figure 4B–D). Vacuoles larger than 2 μm were more frequently detected in XFS group (*p* = 0.183). The cells were loosely connected amongst them and with the basement membrane. Sometimes the epithelium was completely detached from the basement membrane.

Very often there was more than one layer of cells, covering the damaged, underlying cells (Figure 1A,C–E and Figure 3D). The cytoplasmic process that covered the cells could clearly be seen extending from one cell over the neighboring, underlying cell, and it always exhibited a higher density (Figure 1D). Cytoplasm of different cells generally exhibited a wide variety in density (Figure 1).

Nuclei often exhibited an irregular shape, and most commonly they appeared expanded. They often carried dense granules and the chromatin was not uniformly distributed (Figure 3B,C). In many samples, apoptotic bodies were detected (Figure 3D). Very often no nuclei were detected within the cells. Taking into consideration the small size of the lens epithelial cell compared to a normal nucleus and the fact that we studied sequential sections where again no nuclei were observed, we suppose that the nucleus was probably completely missing. The rough endoplasmic reticulum sometimes appeared dilated and the Golgi apparatus and mitochondria were very often swollen (Figure 1).

The cell membranes, especially on their apical side, were often ruptured and lost their regularity (Figure 2). Sometimes there were no cells attached to the basement membrane, or they appeared to have been completely destroyed (Figure 2). It was a common finding that cells lost their regular, cubic shape. Often in XFS patients, the apical cell membrane appeared with varying distances from the basement membrane, due to different cell “heights” (*p* = 0.012) (Figure 1, Figure 2, Figure 3 and Figure 4). The free epithelial cell surface, as a result of these alterations, exhibited an irregular margin (Figure 2 and Figure 4), a characteristic observed in XFS patients (*p* = 0.035).

## 4. Discussion

### 4.1. Cataractic Capsules in Electron Microscopy

The lens capsule in senile cataract lenses is sometimes described as similar to that found in normal lenses [34], but this is not typical. In 1980, Jensen and Laursen reported the presence of intercellular and intracellular vacuoles in lens epithelium insenile cataracts [6]. Nuclei appearing with irregular shapes due to compression of epithelial vacuoles have been described previously [6,35,36,37]. Exo- or endocytotic vesicles were found adjacent to the capsular cells and at the cortical side of the epithelial cells [38]. In accordance with the above-described observations, in our study, intracellular and intercellular vacuoles, influencing the appearance of both the nucleus and the whole cell, were detected in all patients. Additionally, diffuse intracellular oedema was observed in the majority of LECs. These lesions were occasionally so severe and extensive that the cell practically disappeared, a finding not described before.

In an optical microscope study, which is less detailed and accurate than TEM, an abnormal migration and multilayering of LECs was reported in senile lenses [39]. Similarly, in 1989, Bleckmann et al. mentioned a multilamellar arrangement of the LECs, which was an optical microscope finding, even though electron microscopy was also included in their study [40]. To our knowledge, there is no similar description with an electron microscope. We observed cytoplasmic processes covering neighboring, underlying cells. The cytoplasm of these cells always exhibited a lower density, mostly because of an extended diffuse, intracellular oedema, and appeared damaged. There seems to be an attempt of the adjacent healthier cells to protect the unhealthy or compensate for their possible imminent cell death. This theory is in accordance with Charakidas’ et al. suggestion that there are gaps between epithelial cells which are likely to be replaced through cell proliferation at the germinative zone of the anterior lens capsule [41].

A loss of cell membrane integrity was reported using a fluorescent stain for DNA, BOBO-3, in cataractic capsules [42]. This is a finding well established by our observations. Moreover, in a recent TEM study [2], the apical surface of the anterior LECs from patients with different types of cataract was described as smooth. On the other hand, in our study the loss of cells’ regular, cubic shape and sometimes their absence or complete destruction caused an irregular margin of the epithelium. Specifically in XFS patients, this irregularity was mademore distinct by the varying “heights” of the cells.

There are a few studies in which degeneration and transdifferentiation of cataractic lens epithelial cells were observed and discussed. Synder et al. reported that some of LECs showed morphological and immunohistochemical characteristics of mesenchymal cells [37]. In another study, the LECs of anterior subcapsular cataracts were transdifferentiated into spindle-shaped fibroblast-like cells, as demonstrated immunohistochemically. The researchers concluded that the cells may possess bipotential nature and may transdifferentiate into mesenchymal cells. It was suggested that this transdifferentiation may be etiologically connected to the development of cataract [43]. There have been TEM studies commenting that normal appearing epithelial cells were intermingled with abnormal [7,44]. This coexistence of healthy and damaged cells is also demonstrated in the present study, although without visible indications of transdifferentiation.

### 4.2. Special Situations Known to Cause Alterations to Epithelial Cells

In specialized types of cataract, such as intumescent white cataract [9,45], traumatic white cataract [29], electric cataract [27], and atopic cataract [33], alterations of the LECs have been described. These alterations included degenerative lesions, such as swelling, flattening, or loss of cells and nuclear pyknosis or swelling. Patients with these types of cataract were excluded from the present study. The findings in our samples were comparable to those described in the above situations, or were often even more severe.

Straatsma et al. included an anterior lens capsule’s electron micrograph of a patient with choroidal metastatic tumor, after radiation treatment and chemotherapy. The abnormalities of this epithelium were comparable to our specimens, even though none of our patients were exposed to radiation [7]. Moreover, in an experimental study with ultraviolet B (UVB) radiation exposure in mice, lens epithelial cell damage was observed in TEM and there were descriptions of apoptotic cells, apoptotic bodies, nuclear chromatin condensation, throughout the sections of the whole anterior lens surface [24]. Apoptosis was also obvious in our sample, which consists of patients without any specific radiation exposure, except for sunlight.

Diseases like uveitis [8,31], retinitis pigmentosa [26], Wilson’s disease [25], and Alport syndrome [28] have been associated with the presence of lens epithelial lesions, such as holes, thinning and degradation of the epithelium [26], vacuoles [31], capsular dehiscences [28] or granular deposits [25]. This is, therefore, why patients with those diseases were excluded from the study; nevertheless, our observations exhibited similarities with the above lesions.

Diabetes mellitus has also been associated with anterior lens capsule abnormalities. Lozano-Alcazar et al. noted the presence of stratification of the epithelium at the anterior pole of capsules from diabetic patients with senile cataracts [46]. In another study using phosphorous EDXA (Energy-Dispersive X-ray Analysis) a cell density decrease was noticed in diabetic LECs of cataract patients [47]. In the present study, there was no stratification of the epithelium as it was described by Lozano-Alcazar, and the ultrastructural lesions of the LECs did not differ in patients with and without diabetes mellitus.

Overall, the severe alterations described in the specific situationsabove, which were excluded from the present study, are comparable to the lesions observed in our sample.

### 4.3. Sunlight and Cataract

The sunlight exposure has been connected with senile cataracts in most epidemiological studies [48,49,50]. In Greece, Theodoropoulou et al. also correlated sunlight exposure with increased risk for cataracts [51]. Mediterranean countries are known to have sunny weather throughout the whole year, and especially around summer time. Locals are exposed to a greater amount of solar radiation, and for a longer period of time, than people in northern countries, where lens epithelia are usually studied.

Considering, firstly, the fact that UVB radiation is shown to cause lens epithelial cell damage [24], and secondly, the higher UVB radiation exposure in Greece, as reported by the World Health Organization [52], we speculate that this could be the reason for the more intense damage observed in our study.

Apart from the possible environmental risk factor of increased sunlight exposure, there could be additional genetic or epigenetic factors influencing the appearance or severity of senile cataracts in our Greek population sample. There are genetic studies in other countries which correlate the age-related cataract with gene polymorphisms [53,54,55,56] or methylation of DNA [55]. It is possible that differences in genomes or gene expression in different populations may also be responsible for a different ultrastructural appearance of LECs.

## 5. Conclusions

Comparing the findings between XFS and non XFS patients, the observed lesions of the aLCs were more extended and more frequently noticed in the exfoliation group, with the presence of irregular margins and the varying heights of lens epithelium differing to a statistically significant level.

In the present study of Greek patients, the lesionsobserved wereoverall more extended and severe compared to that which have already been described in age-related cataracts. Environmental factors, such as increased ultraviolet B (UVB) radiation exposure in Mediterranean countries, genetic factors, epigenetic factors, or all of them, could contribute to these alterations. Further epidemiological and molecular biology research, such as proteomic and genomic analyses, are needed to justify these results.

## Figures and Tables

**Figure 1 medicina-55-00235-f001:**
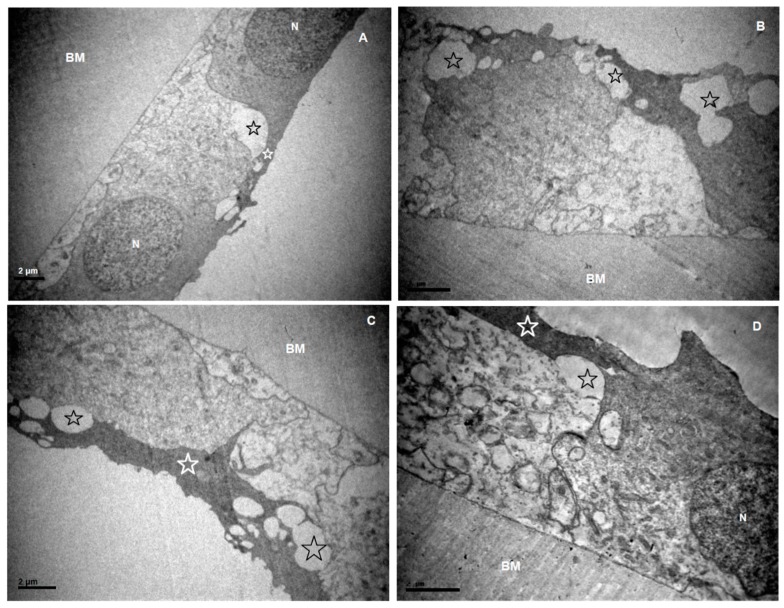
Micrographs of anterior lens capsules (aLCs), with features of oedema. (**A**) ×6000, (**B**) ×8000 and (**C**) ×8000 from the ARC group. (**D**) ×10,000, (**E**) ×6000 and (**F**) ×8000 from the XFS group. (**A**) Features of mild, diffuse intracellular oedema, more obvious in the cell below. (B–D) Neighboring lens epithelial cells (LECs)with features of intense, diffuse intracellular oedema and smaller vacuoles. (**C**,**D**) Electron-denser cytoplasmic process covers the adjacent, more damaged cell. (**E**,**F**) More damaged cells with larger vacuoles, without nuclei.

**Figure 2 medicina-55-00235-f002:**
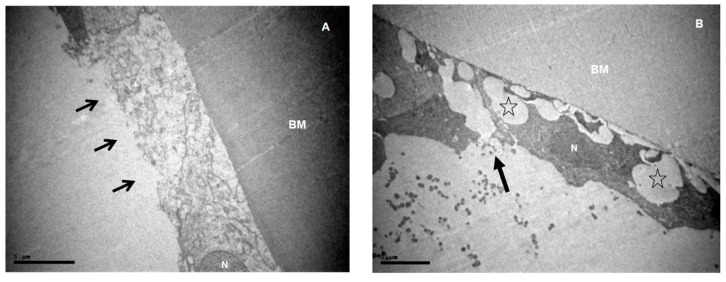
Micrographs of aLCs, with abnormalities of the apical surface membrane (**A**) ×5000 from the ARC group. (**B**) ×4000, (**C**) ×10,000and (**D**) ×5000 from XFS group. (**A**) The only case from the ARC group in which a loss of apical cell membrane is observed. Features of diffuse intracellular oedema are present and no nucleus is detected. (**B**,**C**) Ruptures of the apical cell membrane. (**C**,**D**) Absence of cells in this part of epithelium. (**B**–**D**) The apical LEC’s surface exhibits an irregular margin.

**Figure 3 medicina-55-00235-f003:**
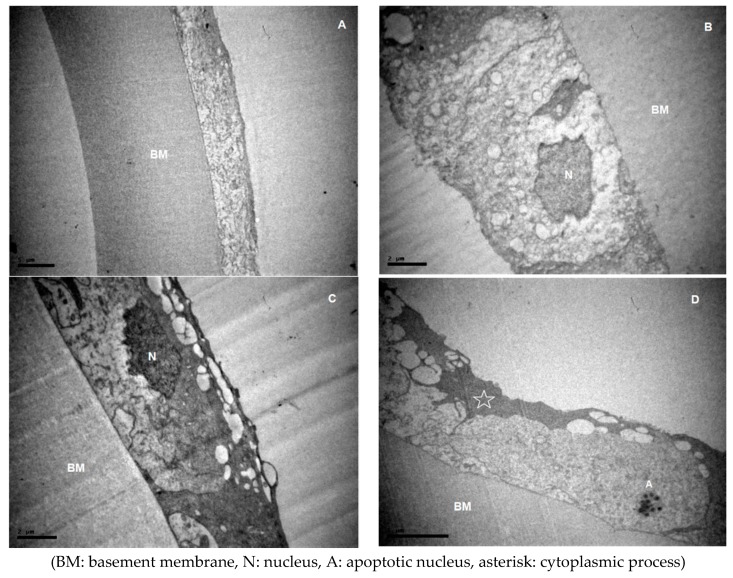
Micrographs of aLCs with abnormalities of nuclei. (**A**) ×3000 from the ARC group. (**B**) ×8000, (**C**) ×8000 and (**D**) ×5000 from the XFS group. (**A**) No nuclei observed in the cells. Features of extensive diffuse intracellular oedema are present. (**B**,**C**) Irregularly shaped nuclei. (**D**) An apoptotic nucleus is present. An electron-denser cytoplasmic process covers the adjacent cell.

**Figure 4 medicina-55-00235-f004:**
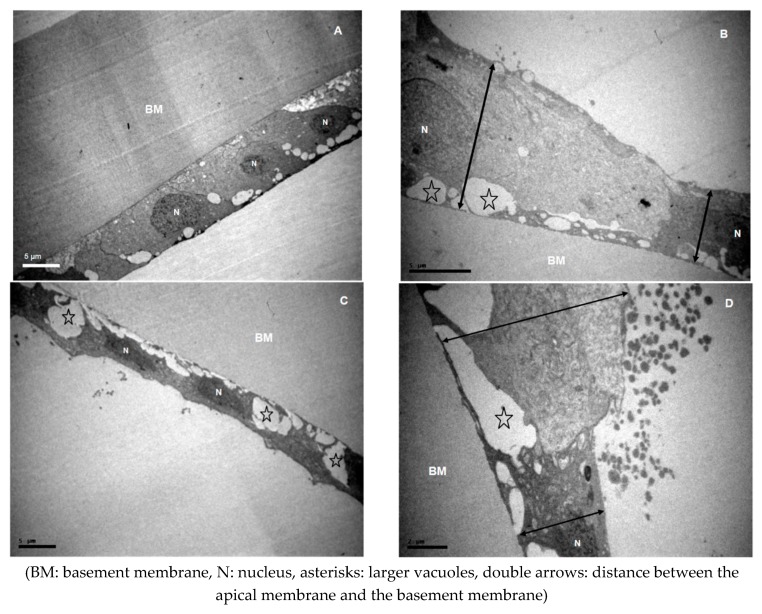
Micrographs of ALCs with variations in cell “heights” and epithelial margin (**A**) ×3000 from the ARC group. (**B**) ×5000, (**C**) ×3000 and (**D**) ×8000 from the XFS group. (**A**) LECs with normally presented nuclei, small vacuoles and a regular epithelial margin. (**B**,**D**) Cells exhibit a great variety in “heights”, creating an irregular margin of the apical LECs’ surface. (**C**) Epithelial cells appear flattened with big vacuoles and reduced distances between the apical membrane and the basement membrane.

**Table 1 medicina-55-00235-t001:** Demographic and statistical data.

Age (Years) (Mean ± SD)	74.57 ± 6.55
Gender	
Male	7 (33.3%)
Female	14 (66.7%)
Diabetes mellitus	6 (28.6%)
Glaucoma	3 (14.3%)

**Table 2 medicina-55-00235-t002:** Analysis between senile cataract patients with and without exfoliation syndrome (XFS). (ARC group: age-related cataract group without XFS).

	XFS Group	ARC Group	*p*-Value
Age (years) (mean ± SD)	76.2 ± 5.8	72.8 ± 7.2	0.247
Gender			
Male	3	4	
Female	8	6	0.659
Diabetes Mellitus	4	2	0.635
Glaucoma	3	0	0.214
IOP (mmHg) (mean ± SD)	16.3 ± 2.8	15.3 ± 2.3	0.385

**Table 3 medicina-55-00235-t003:** Analysis of ultrastructural lesions between XFS and age-related cataract (ARC) group.

	XFS Group	ARC Group	*p*-Value
Diffuse intracellular oedema	7	6	1
Small vacuoles (≤2 μm)	2	5	0.183
Large vacuoles (>2 μm)	9	5	0.183
Multilayering	4	3	1
Absence of nuclei	7	2	0.08
Absent or completely destroyed cells	4	1	0.311
Ruptures of cell membrane	5	1	0.149
Irregular margin	5	0	0.035 (<0.05)
Heights	6	0	0.012 (<0.05)

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
