# Peer review of "Severe Abnormalities of Lens Epithelial Cells in Exfoliation Syndrome: A Transmission Electron Microscopy Study of Patients with Age-Related Cataract"

_medicina, 2019, doi:10.3390/medicina55060235_

Round 1

Reviewer 1 Report

This manuscript describes the characterization of epithelial cells from human age-cataract lens capsules (with or without exfoliation syndrome (XFS)) by transmission electron microscopy. A total of 21 lens capsules from age-related cataract (ARC) patients who had undergo phacoemulsification surgery were examined, out of which 11 also exhibited characteristics of exfoliation syndrome.  Abnormal cellular morphology, particularly characterized by the presence of large vacuoles, was observed in cells from the ARC group, and these were found to be severe in those with XFS.  Overall, this is a straightforward study comparing ARC lens capsule cell morphology to that of XFS lens capsule cells, but the manuscript has major presentation drawbacks.

Some glaring shortcomings of the presentation/description of the data are outlined below.

1.    The age-related cataract samples without XFS are referred to as “control”.  These are not non-cataractous lens samples to refer them to as such.  Ideally, non-cataractous lens anterior capsular tissue should be used as a real control.  Also, it should be clearly stated as an objective that the study represents a comparative account of ARC vs. XFS lens capsular epithelial cell characteristics.

2.    One cannot observe oedema (abnormal accumulation of fluid) by TEM, one can only interpret it.  Therefore these findings should be described as “features” of oedema were observed.

3.    In supplementary material, C1 image on the left is exactly the same as that shown in Figure 1, image on the left, top (Fig. 1A), with the sole exception that it is clockwise rotated 90 degrees.  There are other such examples (For example, C1 right image is a slightly different area of the same image in Figure 1 left bottom image (Fig. 1C)).  One cannot show the same figure twice, once in the “real” manuscript and then again in the supplementary material.

4.    How many cells or areas of each specimen were observed/analyzed by TEM?  This should be described in detail.

5.    If the sections were 1-3 micrometer thin (as described in the Methods), could the absence of the nucleus in some cells simply be described because of the different plane that the cell finds itself in?

6.    Again, because the the sections were 1-3 micrometer thin, how is it concluded that cytoplasmic processes of one cell covers another cell?  Could it be that the cytoplasmic process is abnormal in different areas within the same cell?

7.    It is not clear from figure legends which figures describe ARC vs. XFS samples (Figures 3, 5?). The presentation of the data should be more systematic and should have dedicated figures to make specific points.  For example, first, a general cellular comparisons between the “representatives” of the two groups (in the same figure), followed by specific comparisons such as difference is morphologies in the nucleus, cytoplasm, etc. (Each figure should show the ARC vs. XFS side by side to separately make the above points).

8.    Authors should discuss in more detail how exactly the present study differs from their other published report on the same topic (Sorkou, K.N.; Manthou, M.E.; Tsaousis, K.T.; Brazitikos, P.; Tsinopoulos, I.T. Transmission electron microscopy study of undescribed material at the anterior lens capsule in exfoliation syndrome. Graefe's archive for clinical and experimental ophthalmology = Albrecht von Graefes Archiv fur klinische und experimentelle Ophthalmologie 2018, 256, 1631-1637, doi:10.1007/s00417-018-4062-1).

9.    There are error bars missing in Control Group C1, C8, C10; XFS Group X1, X4 (right image), X8 (right image).

10. “We suggest that environmental factors, such as increased ultraviolet B (UVB) radiation 32 exposure in Mediterranean countries, genetic, epigenetic factors or all of them, could 33 contribute in these alterations.” This conclusion is far-fetched.  To make such a claim, the study should have a comparative account of hundreds of samples from Greek vs. non-Greek (non-Mediterranean, etc.) populations.  This claim should be toned-down.

11. The cell “heights” are found to different.  It is better to describe this as width.

Author Response

Dear Reviewer,

We would like to thank you for your time and effort, your valuable comments and useful remarks.

We made the following changes in the paper according to the suggestions:

1. The age-related cataract samples without XFS are referred to as “control”.” These are not non-cataractous lens samples to refer them to as such.  Ideally, non-cataractous lens anterior capsular tissue should be used as a real control.  Also, it should be clearly stated as an objective that the study represents a comparative account of ARC vs. XFS lens capsular epithelial cell characteristics.

We refrained from using the term “control”, as correctly commented, for the age-related cataract patients without XFS. Instead, we used the term ARC, as suggested.

2. One cannot observe oedema (abnormal accumulation of fluid) by TEM, one can only interpret it.  Therefore these findings should be described as “features” of oedema were observed.

“Features of oedema” replaced “oedema”, which indeed cannot be directly observed.

3.In supplementary material, C1 image on the left is exactly the same as that shown in Figure 1, image on the left, top (Fig. 1A), with the sole exception that it is clockwise rotated 90 degrees.  There are other such examples (For example, C1 right image is a slightly different area of the same image in Figure 1 left bottom image (Fig. 1C)).  One cannot show the same figure twice, once in the “real” manuscript and then again in the supplementary material.

Thank you for your thorough observation. The supplementary material was improved according to the suggestions. The purpose of this material was to demonstrate the most representative micrographs of each case. In 3 cases we inadvertently included same or similar micrographs to the ones from the manuscript. This has been taken care of.

4.How many cells or areas of each specimen were observed/analyzed by TEM?  This should be described in detail.

Information was added about the areas of cells used for observation in the study and about the golden colour of the utrathin sections, which verifies we have the proper section for the best observation.

5.If the sections were 1-3 micrometer thin (as described in the Methods), could the absence of the nucleus in some cells simply be described because of the different plane that the cell finds itself in?

Very often no nuclei were detected within the cells. To explain this we added this in the text: “Taken into consideration the small size of the lens epithelial cell compared to a normal nucleus and the fact that we studied sequential sections where again no nuclei were observed, we suppose that the nucleus is probably completely missing. It is possible that the cell is apoptotic.”

We would like to add the following thought: The space between the cell nucleus and the cell membrane is supposed to be very small, normally 1-2 μm. The possibility that our ultrathin section (which is maximum 1 μm thickness) is taken perpendicularly from that exact space is rather small. Moreover, and most importantly, we took sequential sections where nuclei were again not visible.

6.Again, because the the sections were 1-3 micrometer thin, how is it concluded that cytoplasmic processes of one cell covers another cell?  Could it be that the cytoplasmic process is abnormal in different areas within the same cell?

Regardless of the thickness of the section, the cytoplasmic process that covered the cells could clearly be seen extending from one cell over the neighboring, underlying cells. The origin of the cytoplasmic process was clearly seen during microscopy and its cell membrane is in continuation with the cell membrane of the rest of the cell. In figures 1A and 1D for example, the micrographs were focused between the 2 neighboring cells in order to demonstrate that the cell on the right “sends” a cytoplasmic process over the cell on the left (marked with a white asterisk).

7.It is not clear from figure legends which figures describe ARC vs. XFS samples (Figures 3, 5?). The presentation of the data should be more systematic and should have dedicated figures to make specific points.  For example, first, a general cellular comparisons between the “representatives” of the two groups (in the same figure), followed by specific comparisons such as difference is morphologies in the nucleus, cytoplasm, etc. (Each figure should show the ARC vs. XFS side by side to separately make the above points).

Thank you for the remark about our presentation of photographs. All figures now include photos from both groups of patients, which are noted on the legends. Each figure is focused on a type of finding which is mainly presented (Figure 1 is mostly about features of oedema, Figure 2 is about the abnormalities of the apical surface membrane, Figure 3 is about abnormalities of nuclei and Figure 4 is about epithelial margin) It is expected that all the described abnormalities in this study are not found individually in each case. Instead, each case presents with a variety of features.

8.Authors should discuss in more detail how exactly the present study differs from their other published report on the same topic (Sorkou, K.N.; Manthou, M.E.; Tsaousis, K.T.; Brazitikos, P.; Tsinopoulos, I.T. Transmission electron microscopy study of undescribed material at the anterior lens capsule in exfoliation syndrome. Graefe's archive for clinical and experimental ophthalmology = Albrecht von Graefes Archiv fur klinische und experimentelle Ophthalmologie 2018, 256, 1631-1637, doi:10.1007/s00417-018-4062-1).

More detail was provided about the published report on the same topic, which focused on the findings in the subepithelial region of aLC, towards the lens fibers. The study was about the presence of a new, not previously described, unbound material, which consisted of electron-dense microgranules or larger formations, on the apical side of lens epithelium in XFS group

9.There are error bars missing in Control Group C1, C8, C10; XFS Group X1, X4 (right image), X8 (right image).

They were added, as requested.

10.We suggest that environmental factors, such as increased ultraviolet B (UVB) radiation 32 exposure in Mediterranean countries, genetic, epigenetic factors or all of them, could 33 contribute in these alterations.” This conclusion is far-fetched.  To make such a claim, the study should have a comparative account of hundreds of samples from Greek vs. non-Greek (non-Mediterranean, etc.) populations.  This claim should be toned-down.

We toned down the statement about the effect of the environmental factors, as suggested. We tried to present it as a reasonable hypothesis which needs further investigation, in both abstract and the conclusion.

11.The cell “heights” are found to different.  It is better to describe this as width.

The distance between the beginning of the basement membrane and the LECs’ free surface in histology is referred to as “height” of the cell, but we tried to describe it in a clearer way, so it is not misunderstood. The term height was used in brackets.

In addition,

Improvements were made in the abstract, according to the suggestions

The introduction was improved. We provided additional backround information

The presentation of the results, tables and figures, were corrected as suggested.

Sincerely yours,

Maria Eleni Manthou, MD, PhD

Reviewer 2 Report

In the  article entitled " Severe abnormalities of lens epithelial cells in exfoliation syndrome: a transmission electron microscopy study of patients with age related cataract", authors made a good attempt to explain the differences between abnormalities associated with lens epithelial cells in exfoliation syndrome using transmission electron microscope. Authors presented a good view of the of difference between lens epithelial cells in patients with exfoliation and without exfoliation syndrome.  It would be more helpful if authors could make an attempt to explore the proteomics or genomics of the same tissue which could be beneficial in developing the therapeutic approaches. Over all the article has been presented very well.

Author Response

Dear Reviewer,

We would like to thank you for your time and effort, your positive comments and useful remark.

1. We made an attempt to improve the manuscript with small changes and we added some information in the methodology.

2. Application of proteomics or genomics in our specimen was not included in our initial design of the study. After having these ultrastructural results, we agree it will be very interesting to expand our research in these scientific fields. We also added this as a comment in our conclusion.

Sincerely yours,

Maria Eleni Manthou, MD, PhD

Round 2

Reviewer 1 Report

Authors have addressed previous concerns.

Author Response

Dear Reviewer,

We would once again like to thank you for your time and effort to help us improve our manuscript.

We asked for professional help to improve the English writting. All changes are again marked with yellow.

Sincerely yours,

Maria Eleni Manthou, MD, PhD
